# In Vivo Oral Sentinel Lymph Node Mapping by Near-Infrared Fluorescent Methylene Blue in Rats

**DOI:** 10.3390/diagnostics12112574

**Published:** 2022-10-24

**Authors:** Yu-Xiao Wu, Qian-Ying Mao, Yi-Fan Kang, Shang Xie, Xiao-Feng Shan, Zhi-Gang Cai

**Affiliations:** 1Department of Oral and Maxillofacial Surgery, Peking University School and Hospital of Stomatology, Beijing 100081, China; 2National Center of Stomatology, Beijing 100081, China; 3National Clinical Research Center for Oral Diseases, Beijing 100081, China; 4National Engineering Research Center of Oral Biomaterials and Digital Medical Devices, Beijing 100081, China

**Keywords:** methylene blue, near-infrared fluorescence, sentinel lymph node, mapping, indocyanine green

## Abstract

This study aimed to demonstrate the feasibility of near-infrared (NIR) fluorescence imaging using methylene blue (MB) for detecting oral sentinel lymph nodes (SLNs) in rats and compared MB’s tracer effects with those of indocyanine green (ICG) in SLN mapping. Different concentrations of MB were injected into the rats’ left lingual submucosa to determine the optimal concentration by using a continuous (1 h) MI-1 fluorescence imaging system. To compare the tracer effects of the optimal MB concentration with ICG in oral SLN mapping, MI-1 imaging was continuously monitored for 12 h. The mean signal-to-background ratio (SBR) of the SLNs and SLN fluorescence area fraction were analyzed. SLNs and lymphatic vessels were clearly visible in all rats. The optimal injection dose of MB infected into lingual submucosa for NIR fluorescence imaging was 0.2 mL of 6.68 mM MB. During continuous monitoring for 12 h, the mean SBR of the SLNs was significantly higher in the ICG groups than in the MB groups (*p* < 0.001). However, the area fraction of SLN fluorescence in the ICG groups increased continuously, owing to strong fluorescent contamination. This study examined the feasibility of detection of draining lymph nodes in the oral cavity of rats using MB NIR fluorescence imaging. MB causes less fluorescent contamination than does ICG, which shows promise for clinical research and application.

## 1. Introduction

Elective neck dissection (END) at the time of primary tumor resection affords a survival benefit in patients with clinically node-negative (cN0) oral squamous cell carcinoma [1]. However, only 20–40% of patients with cN0 oral squamous cell carcinoma develop occult cervical metastasis, which means there is a risk of unnecessary neck dissection in 60–80% of patients who have truly negative cervical lymph nodes [2]. Therefore, efforts are being made to determine a method to accurately identify and predict occult cervical metastasis and adapt to the trend of minimally invasive tumor surgery.

Sentinel lymph node biopsy (SLNB) is the diagnosis and treatment standard for malignant melanoma or breast cancer [3,4]. Patients with SLN metastasis must undergo a completion neck dissection, while those without SLN metastasis can be further followed up [5]. The main tracers for detecting SLNs in head and neck mucosal cancers are blue dyes and radioactive colloids [6,7]. However, SLNs labeled with blue dyes can only be detected after surgery, while nuclear medicine and equipment are needed for the use of radioactive colloids. Furthermore, the hazards of radionuclides should be carefully considered [8,9,10,11].

Recently, near-infrared (NIR) fluorescence imaging has gained popularity because it allows real-time and noninvasive detection of deep tissues [12,13]. In patients with oral cancer, NIR fluorescence imaging with indocyanine green (ICG) has been successfully used to detect SLN; however, particular caution needs to be exercised in the case of patients who have a history of allergy to iodides [2,8,14,15]. Methylene blue (MB) has been used not only as a staining reagent but also as a NIR fluorescent material, emitting fluorescence at 700 nm [16]. Furthermore, this fluorescent dye, which is affordable and accessible and has low toxicity, helps to detect exposed SLN synchronously because of the blue color and fluorescence [11,17,18,19]. NIR fluorescence imaging using MB has been used to identify ureteral location [16,20,21,22,23]. However, no animal studies have been conducted to assess the potential for clinical use of MB NIR fluorescence imaging for oral SLN mapping.

Therefore, in this study, we aimed to examine the feasibility of NIR fluorescence imaging using MB for the detection of oral SLNs and associated lymphatic vessels in rats. To this end, we used oral lymphatic drainage in rats to mimic SLN identification in oral cancer patients. Furthermore, we determined the optimal concentration of MB for NIR fluorescence imaging and compared the effectiveness of this approach with that of tracking SLN with ICG to provide a basis for clinical research and applications of NIR fluorescence imaging with MB for SLN mapping in oral cancer.

## 2. Materials and Methods

### 2.1. Animal and Dye Preparations

All animal experiments were approved by the Ethics Committee for Animal Research, Peking University Health Science Center (License No. LA2019336) and adhered to the Declaration of Helsinki. Eighty adult male Sprague-Dawley rats (450 to 550 g) were provided by the Laboratory Animal Service Center, Peking University Health Science Center.

MB injection (26.74 mM; Jichuan Pharmaceutical, Ltd., Taizhou, Jiangsu, China) was diluted with 0.9% saline solution to yield 3.34, 6.68, and 13.37 mM MB solutions. ICG (25 mg vials; Dandong Yichuang Pharmaceutical, Ltd., Dandong, Liaoning, China) was mixed with 10 mL of sterile water to yield a solution of 2.5 mg/mL (3.2 mM).

### 2.2. Real-Time Mapping of SLNs via Intraoperative NIR Fluorescence Imaging and Measurement of the Velocity of MB Movement

Fifty rats were randomly divided into the MB and control groups. The animals were anesthetized with an intraperitoneal injection of 2.5% pentobarbital (40 mg/kg) and fixed in the supine position. Hair was removed from the head and neck. The rats in the MB groups were injected with 0.2 mL of different concentrations of MB solution (3.34, 6.68, 13.37, and 26.74 mM, respectively, 10 rats per concentration) into the left lingual submucosa. On the other hand, the animals in the control groups (10 rats) were injected with 0.2 mL of saline solution into the left submucosa. The MI-1 fluorescence vascular imager (Micro Intelligence Technology, Ltd., Jinan, China) was used to continuously monitor the SLNs and related lymphatic vessels for 1 h in each rat, and a series of images were obtained after MB injection. The first transcutaneous visible NIR fluorescent hotspots were considered the SLNs. To determine the optimal injection concentration of MB for NIR fluorescence imaging, the fluorescence intensity (FI) of the SLNs and background intensity were quantified using custom MI-1 software, and the initial and optimal imaging times required to observe SLNs’ fluorescence were recorded. FI was then expressed as the signal-to-background ratio (SBR), defined as the ratio of the fluorescence intensity of the SLNs to the background intensity of the surrounding tissue. An SBR higher than 2 was regarded as clinically relevant [24].

In addition, the distance between the injection site and the SLNs was recorded to measure the movement velocity for each MB concentration. The velocity of MB movement along the lymphatics was assessed by dividing the distance between the injection site and the SLNs by the time from injection to the detection of fluorescence (i.e., initial imaging time). 

The animals were sacrificed by an overdose of pentobarbital after 1 h. All the tissues showing fluorescence were dissected for NIR fluorescence imaging to verify MB uptake and then sent to the pathologist for pathologic identification.

### 2.3. A Comparative Study of MB and ICG

Thirty rats were divided randomly into the MB, ICG, and control groups. Rats were injected with 0.2 mL of different agents (MB, ICG, and saline, respectively, 10 rats per agent) into the left lingual submucosa. The MI-1 fluorescence vascular imager was used to continuously monitor the SLNs and related lymphatic vessels for 12 h in each rat and a series of images was then obtained after the injections. To compare the effect of the optimal concentration of MB for NIR fluorescence imaging on tracking SLN with that of ICG (2.5 mg/mL), the SBR for different agents over time was calculated and the SLN fluorescence area fraction, which represents the SLN fluorescence area on a planar fluorescence image, was measured using image analysis software (Image J 1.32j, NIH, Bethesda, MD, USA) [25]. The rats were sacrificed after 12 h and then the neck space was dissected to study the distribution of the fluorescent agent in SLNs and other tissues.

### 2.4. Statistical Analysis

Data were expressed as mean ± standard deviation (SD) values and analyzed using the SPSS statistical software package (Version 25.0; SPSS, Inc., Chicago, IL, USA). One-way ANOVA with the least significant difference (LSD) multiple comparison test was performed for each assessment parameter to examine statistical differences between three or more groups. The differences in SBR between different groups and time points were tested using repeated-measures ANOVA. For all statistical tests, significance was 2-sided and set to *p* < 0.05, and 95% confidence intervals were used to assess the precision of the obtained estimates.

## 3. Results

### 3.1. Real-Time Mapping of the SLNs via Intraoperative NIR Imaging and Measurement of the Velocity of MB Movement

Compared with the control groups treated with saline, the SLNs and adjacent segment lymphatic vessels in the MB groups exhibited significant brightness. The fluorescence signal of SLNs and lymph vessels increased strongly after MB injection, and over time, the SLNs signal continued to increase and reached a steady state 15 min after injection, while the lymph vessels signal decreased. However, the fluorescence signal of the lowest concentration groups (3.34 mM) appeared within 1 min and then decreased quickly (Figure 1A). 

The mean SBR of the SLNs was 1.94 ± 0.19 (*n* = 10), 4.69 ± 1.23 (*n* = 10), 6.57 ± 1.45 (*n* = 10), 6.30 ± 1.48 (*n* = 10), and 5.01 ± 1.39 (*n* = 10) for the control, 3.34, 6.68, 13.37, and 26.74 mM groups, respectively. A repeated-measures ANOVA showed a significant difference between time points (*p* < 0.001) and concentration groups (*p* < 0.001). The SBR of the 6.68 mM concentration group was higher than that of the other concentrations (*p* < 0.05) (Figure 1B). Increasing the MB concentration from 3.34 to 26.74 mM led to an increase in the SLNs’ initial and optimal imaging times from 0.60 and 1.44 min to 5.05 and 23.20 min, respectively (Table 1). Therefore, the concentration of 6.68 mM was adopted as the optimal concentration of MB for NIR fluorescence imaging based on reliably high SBR, rapid uptake, and good retention.

The median distance between the injection site and the SLNs was 2.5 cm (range: 2.1 to 3 cm). The mean velocity of MB movement was 4.47 ± 1.32, 2.16 ± 0.85, 1.33 ± 0.45, and 0.55 ± 0.21 in the 3.34, 6.68, 13.37, and 26.74 mM groups, respectively. The velocity of MB movement significantly decreased with increased MB concentrations (Figure 1C). Fluorescence could still be observed clearly after incision and exposure of the nodes (Figure 2A). All resected tissues appeared fluorescent when examined under the NIR source and were confirmed to be lymphatic tissue by hematoxylin-eosin staining (Figure 2B,C).

### 3.2. A Comparative Study of MB and ICG

Almost no fluorescence signal was detected in the control groups injected with saline. In rats treated with 6.68 mM (2.5 mg/mL) MB and 2.5 mg/mL ICG, the SLNs and adjacent segment lymphatic vessels were successfully identified after injection, and the fluorescence was sustained for approximately 12 h. However, the fluorescence signal of the SLNs in the MB groups weakened to approximately normal levels at 12 h after injection (Figure 3A).

The mean SBR of the SLNs was 5.70 ± 1.83 (*n* = 10), 8.72 ± 1.93 (*n* = 10), and 1.85 ± 0.20 (*n* = 10) in the MB, ICG, and saline groups, respectively. Repeated-measures ANOVA showed a significant difference between time points (*p* < 0.001) and concentration groups (*p* < 0.001). The mean SBR of the SLNs was significantly higher in the ICG groups than in the MB and saline groups (*p* < 0.001) (Figure 3B). The area fraction of SLN fluorescence in the MB groups increased from 0.03% at 0 h to 1.94% at 1 h, and then gradually decreased to 0.2% at 12 h. At 1 h, the area fraction of SLN fluorescence is comparable to the area fraction of a dissected lymph node as measured with a caliper. However, the area fraction of SLN fluorescence in the ICG groups increased from 0.03% at 0 h to 9.98% at 12 h. At 1 h, the ICG groups had a 1.69% higher area fraction of SLN fluorescence than the MB groups (*p* < 0.001) (Figure 3C). At 12 h after the neck dissection, we found that besides ipsilateral SLNs, the fluorescence signal was also clearly detectable in contralateral SLNs and other surrounding tissues, such as the submandibular gland, indicating that ICG causes fluorescent contamination. Such contamination may stain the operative field, impairing adequate vision. However, the fluorescence signal in the MB groups was almost detected in the ipsilateral SLNs, which were easily distinguishable from other tissues (Figure 4).

## 4. Discussion

This study introduces a transcutaneous technique for SLN mapping after injection of a low concentration of MB. MB has been commonly used as a staining reagent in high concentrations to resect fistulas in surgical procedures [26]. The results of the present study examined the feasibility of identifying SLNs and related lymphatic vessels in the oral cavity of rats by using diluted MB and NIR fluorescence. The SLN could be mapped synchronously by the blue color and NIR fluorescence of MB, which is similar to the combination of blue dyes and radioactive colloids. However, radionuclides are inherently hazardous, while NIR fluorescence is simple, safe, rapid, and sensitive [11]. 

Since clinical and animal information regarding the use of NIR-fluorescent MB for oral SLN detection is currently unavailable, this study aimed to assess the optimal concentration of MB for NIR fluorescence imaging in oral SLN identification. In our study, all MB concentrations showed a mean SBR higher than 2, which is considered adequate to distinguish the target from the background. The mean SBR in the 6.68 mM concentration group was higher than that in the other concentrations. In addition to evaluating the SBR of MB at different concentrations over time, we also recorded the initial and optimal imaging times at each concentration. Due to the high SBR and reliably rapid uptake at 6.68 mM, this concentration was adopted as the optimal concentration of MB for NIR fluorescence imaging, which yielded an initial and optimal imaging time of 1.30 and 15.00 min after injection (velocity, 2.16 cm/min). This rate of MB movement is helpful for rapid recognition of SLN in clinical practice. Furthermore, we found that the concentration of MB affects the rate of migration of methylene blue molecules. Specifically, with an increased MB concentration, the SLN fluorescence initial and optimal imaging times increased, while the velocity of MB movement decreased. This could be because this small molecule is less adhesive to plasma proteins. The high concentration can be exposed to more plasma proteins, so that the elimination of MB is slow, and the effect is maintained for a long time, whereas the low concentration can be transported more quickly throughout the lymphatic vessels to the SLN after injection.

ICG has been successfully used to detect SLN in oral cancer, and ICG-based identification shows potential benefits and drawbacks [2,3,14,15,27,28]. In the present study, the mean SBR of SLNs was significantly higher in the ICG groups than in the MB groups, and the fluorescence signal of the SLNs was still apparent 12 h after injection in the ICG groups. The strong ICG fluorescence could still be visualized even 12 h after injection in rats. In dogs, lymph nodes have been reported to show residual fluorescence at 24 h after ICG solution injection [29]. Although fluorescence was also sustained for 12 h after MB injection at the optimal concentration, the fluorescence signal of the SLNs weakened 12 h after MB injection in our study. In rabbits, the fluorescence of MB in the lymph node was reported to be retained for more than 2.5 h [11].

We calculated the SLN fluorescence area fraction to evaluate the tracer effect of different agents. Over 12 h of continuous monitoring, the area fraction of the SLN fluorescence in the MB groups was 1.94% at 1 h, comparable to the area fraction of a dissected lymph node as measured with a caliper, and the area fraction then gradually decreased to near pre-injection levels. Based on these findings, we proposed the optimal period for SLN detection (15 min to 1 h after injection) with high SBR and appropriate fluorescence at an optimal concentration of MB in NIR fluorescence imaging. Assessments during this period allow the surgeon to confirm that all the SLNs have been resected. In contrast, the area fraction of SLN fluorescence increased over time from 0 to 12 h in the ICG groups. Uptake of ICG by other tissues such as the submandibular gland was noted 12 h after neck dissection. Since the submandibular gland does not represent lymphatic tissue, these results indicate that ICG can also be absorbed by the submandibular gland and can easily bind to plasma proteins, causing fluorescent contamination and staining the operative field, thereby impairing adequate vision. However, due to its lower adhesion and higher absorption, MB causes less fluorescent contamination than ICG, making it suitable for observation.

To the authors’ knowledge, mapping of SLN through NIR fluorescence of MB for oral cancer has not been attempted before. There are other significant potential benefits in this method. MB is relatively accessible and economical, making fluorescence vascular imaging relatively cheap [11]. Other low-cost blue dyes such as Evans blue and patent blue, which are also useful choices for SLN mapping, also show NIR fluorescence [29]. However, patent blue belongs to the group of triarylmethane synthetic dyes, which can cause higher hypersensitivity reactions, whereas MB rarely induces hypersensitivity and is generally regarded as a safer alternative to triarylmethane dyes for lymph node mapping [30]. MB also contains no iodine, unlike ICG, which contains less than 5% iodine and can induce severe adverse reactions in patients sensitive to iodide [31]. The proposed approach can be used for transcutaneous detection of SLN and may allow visualization of lymphatic vessels, which may be attributed to deeper draining lymph nodes. We believe that NIR fluorescence imaging detection of SLN can be used as an intraoperative imaging technique and this MB fluorescence tracer is of great value to clinical research.

We acknowledge several limitations of this study. First, this study only examined the effect of the concentration of MB on rate, and other factors that may affect the rate of migration of methylene blue molecules, including body position, injection position, direction and volume, and post-injection massage, were not assessed. Thus, clinicians should consider the potential effect of the above factors affecting SLN uptake after injection of MB solution in clinical cases. In addition, we were not able to consider deeper fluorescence penetration, which made it challenging to identify SLN during surgery. Thus, further studies are needed to explore the maximum penetration depth when applying transcutaneous NIR fluorescence imaging after injection of MB to better apply in clinical practices. Third, the main objective of our study was intended to provide information regarding the timing and efficacy of this imaging technique, and we selected a healthy rat model to mimic oral SLN identification without real tumor metastases. However, the lymphatic drainage pattern of tumors is complex, and we plan to evaluate the diagnostic performance in tumor models before attempting clinical validation in patients with early oral cancer.

## 5. Conclusions

This study examined the feasibility of identification of SLNs and related lymphatic vessels in the oral cavity of rats by using dilute MB and NIR fluorescence. The optimal MB concentration in 0.2 mL of solution injected into lingual submucosa was 6.68 mM and the optimal interval for SLN detection was 15 min to 1 h after injection. This period allows the surgeon to confirm that all the SLNs have been resected. MB causes less fluorescent contamination than ICG, which is suitable for observation. Consequently, NIR fluorescence imaging using MB shows great promise for clinical research and application.

## Figures and Tables

**Figure 1 diagnostics-12-02574-f001:**
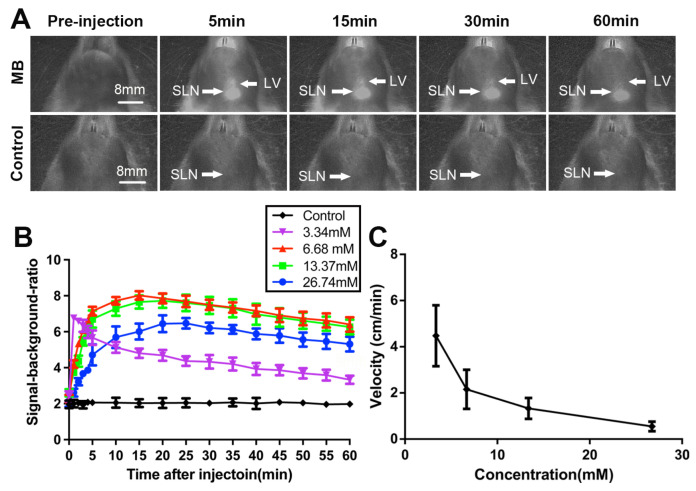
Sentinel lymph node mapping using MB NIR fluorescence imaging in rats. (**A**) Fluorescence images acquired before injection and 5, 15, 30, and 60 min after the injection of 6.68 mM MB and saline in rats. SLN = sentinel lymph node, LV = lymphatic vessel. (**B**) Changes in the SBR of the concentration groups (mean ± SD) within the SLN region over time after injection. Differences were observed between concentration groups (*p* < 0.001). (**C**) The velocity of MB movement (mean ± SD) is plotted as a function of the MB concentration. The velocity of the MB movement significantly decreased with the increased MB concentration (*p* < 0.001).

**Figure 2 diagnostics-12-02574-f002:**
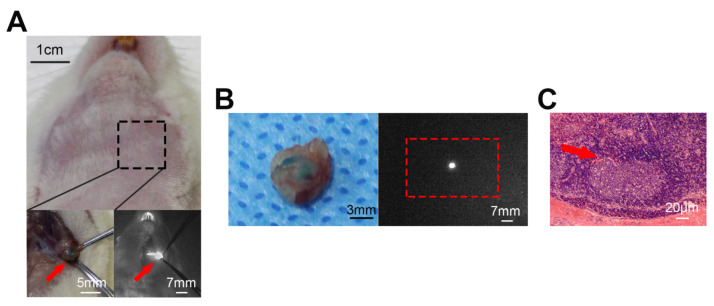
Inspections of the surgical field and the resected SLN. (**A**) Photographs of a rat before dissection (**top**), neck region after dissection (**bottom left**), and fluorescence observation after dissection (**bottom right**). (**B**) The same SLN after resection from the above rat. Shown are the bright-view image (**left**) and NIR fluorescence image (**right**). (**C**) Hematoxylin-eosin staining of the SLN (40× magnification). The arrow shows a germinal center of the SLN.

**Figure 3 diagnostics-12-02574-f003:**
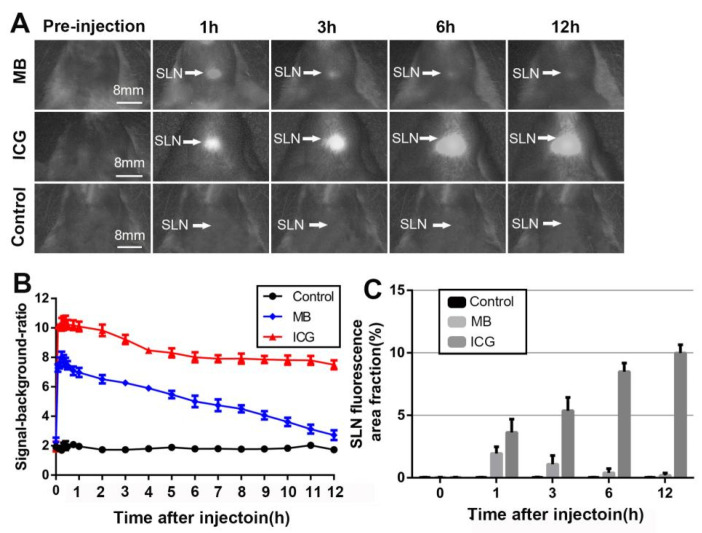
Comparison between MB and ICG for sentinel lymph node mapping using NIR fluorescence imaging in rats. (**A**) Fluorescence images acquired before injection and 1, 3, 6, and 12 h after injection of MB, ICG, and saline in rats. (**B**) Changes in the SBR of different groups (mean ± SD) within the SLN region over time after injection. Differences were observed between different groups (*p* < 0.001). (**C**) SLN fluorescence area fraction before injection and 1, 3, 6, and 12 h after injection of different agents. Values with different agents are significantly different from each other (*p* < 0.001).

**Figure 4 diagnostics-12-02574-f004:**
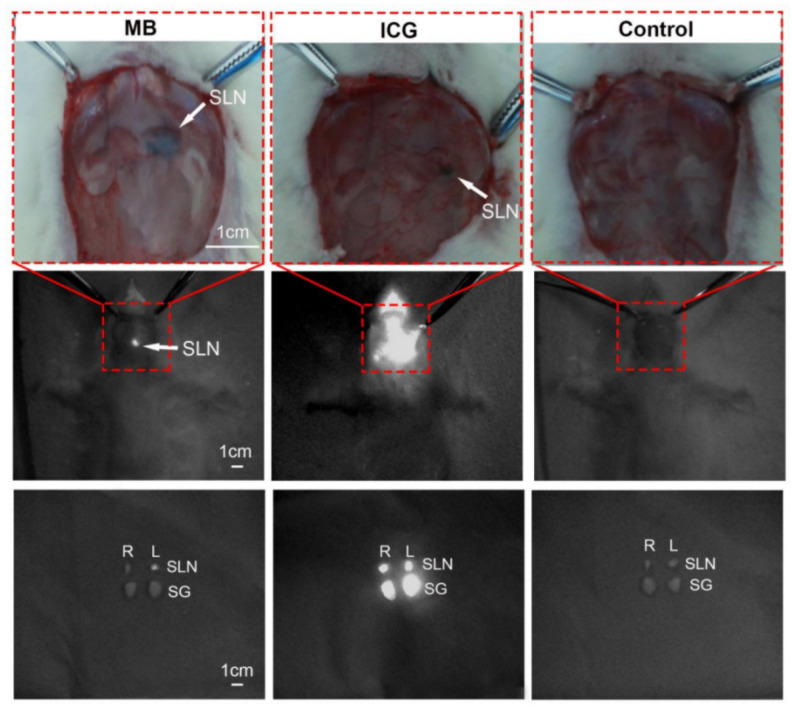
Bright-view images (**top**) and NIR fluorescence images (**middle**) of the neck of rats at 12 h after the injection of MB, ICG, and saline. NIR fluorescence images (**bottom**) of different tissues from the rats at 12 h after injection of MB, ICG, and saline. SLN = sentinel lymph node, SG = salivary gland, R = right, and L = left.

**Table 1 diagnostics-12-02574-t001:** The initial and optimal imaging times required to observe SLN fluorescence in the concentration groups.

Concentration	Sentinel Lymph Node	
	Initial Imaging Time (min)	Optimal Imaging Time (min)
Control	0	0
3.34 mM	0.600:20	1.441:25
6.68 mM	1.300:47	15.001:89
13.37 mM	2.080:53	18.052:48
26.74 mM	5.051:27	23.202:53

## Data Availability

The data presented in this study are available upon request from the corresponding author.

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
