# Peer review of "In Vivo Oral Sentinel Lymph Node Mapping by Near-Infrared Fluorescent Methylene Blue in Rats"

_diagnostics, 2022, doi:10.3390/diagnostics12112574_

Round 1

Reviewer 1 Report

In the manuscript titled “In vivo oral sentinel lymph node mapping by near-infrared fluorescent methylene blue in rats”, authors showed usefulness of methylene blue as a probe for imaging oral sentinel lymph node in rats in vivo. This research paper should be published in Diagnostics. But, before publication, the following points should be added or corrected.

1. Line 18-19. “The optimal concentration of MB for NIR fluorescence imaging was 6.68 mM.” should be shown as “The optimal injection dose of MB infected into lingual submucosa for NIR fluorescence imaging was 0.2 mL of 6.68-mM MB.

2. Figure 1B. I think that SBR in the Figure 1B are not consistent with TEXT (Line 132,133).

3. Figure 1A and 1B. SBR for control is more than 50. Therefore, saline injection allowed imaging SLN and MB is not needed for SLN imaging. Is my understanding correct? Please show arrow indicating SLN in photos for control in Figure 1A.

4. Table 1. “Optical imaging time” should be corrected to “Optimal imaging time”.

5. Figure 2B. What is the red dashed square?

6. Figure 2C. Please show a fluorescence image of adjacent section of H-E stained section.

7. Figure 3B. SBR at 1 h following probe injection are not consistent with SBR in Figure 1B.

8. Figure 3A and 3B. SBR for control is more than 2 in Figure 3B. Therefore, SLN was identified with saline injection. Please show arrow indicating SLN in photos for control in Figure 3A.

9. Line 175-190 are same to Line 191-206. Please correct.

10. Line 234-236. The authors mentioned “This could be because this small molecule does not bind to plasma proteins; thus, the low concentration can be transported more quickly throughout the lymphatic vessels to the SLN after injection.” But, this explanation is not enough to understand the relationship between MB dose and velocity. Please properly explain why high dose of MB showed slow movement.

11. Line 293-294. “The optimal MB concentration was 6.68 mM” should be shown as “The optimal MB concentration in 0.2-mL solution injected into lingual submucosa was 6.68 mM”.

Author Response

Dear  Reviewers,

Thank you very much for your letter and giving us a chance to revise the manuscript. We also would like to thank the reviewers for their kind constructive suggestions to improve the quality of the paper . After carefully studying the comments, we have made corresponding changes. Here, we submit the revised version of our manuscript. All the amendments are highlighted in the revised manuscript.

The replies are as follows.

Reviewer 2 Report

Comments to the manuscript,

Figure 2 part C, image quality can be improved

Line 270 reference 30 is in superscript correct [30]

Author Response

Dear Reviewer,

Thank you very much for your letter and giving us a chance to revise the manuscript. We also would like to thank the reviewers for their kind constructive suggestions to improve the quality of the paper . After carefully studying the comments, we have made corresponding changes. Here, we submit the revised version of our manuscript. All the amendments are highlighted in the revised manuscript.

The replies are as follows.
